# Metabolomic characteristics of aerobic and resistance exercise modes

Junjie Kuang[1], Jie Ju[2], Xin Xu[3]*

1 Women's Healthcare Department, Jiangsu Province Hospital, The First Affiliated Hospital with Nanjing Medical University, Nanjing, Jiangsu, China, 2 School of Physical Education, Shanghai University of Sport, Shanghai, China, 3 Shanghai Anti-doping Laboratory, Shanghai University of Sport, Shanghai, China

* xxu2000@outlook.com

## Abstract

Aerobic and resistance exercises are the two most common modes of physical activity. They may cause some functional changes such as elevation of VO2max and muscle mass. However, descriptions of changes in complex molecular network induced by exercise are often insufficiently comprehensive, limiting the exploration of some new indicators. We utilized a metabolomics analysis method based on Liquid chromatography-mass spectrometry (LC-MS) to investigate the metabolic characteristics of 10 healthy male college students at two time points before and after a single session of aerobic and resistance exercise. The analysis was conducted at both the metabolite and metabolic pathway levels. Notably, the concentrations of several amino acids including aspartic acid, glutamic acid, histidine and tryptophan exhibited significant changes following both modes of exercise. These findings offer a more comprehensive understanding of the molecular effects of acute exercise on the human body, contributing to evaluating post-exercise physiological states and screening for relevant metabolite indicators. Future research could employ multi-omics approaches to validate these results and explore the long-term impact of exercise on human metabolic profiles, linking specific metabolic pathways to functional outcomes.

## 1. Introduction

WHO's Global Status Report on Physical Activity released in 2022 shows that more than 80% of adolescents and 27% of adults do not meet recommended levels of physical activity [1]. Although the benefits of exercise are widely recognized, questions remain regarding how to exercise more effectively [2,3]. Both aerobic and resistance exercise are commonly employed in rehabilitation programs. Aerobic exercise is generally recognized for improving exercise endurance, while resistance exercise enhances muscle size and strength. For healthy individuals, training programs should integrate both aerobic and resistance exercises to achieve comprehensive functional

**Data availability statement:** All relevant data are within the manuscript and its Supporting Information files.

**Funding:** The author(s) received no specific funding for this work.

**Competing interests:** The authors have declared that no competing interests exist.

improvement [4]. However, for different medical conditions or specific exercise requirements, the selection of exercise mode must be more targeted. Exercise prescriptions must be tailored to meet individual needs [5,6].

In clinical research, several functional indicators are commonly employed to evaluate the efficacy of exercise, while the exploration of their physiological mechanisms is still in progress. [7,8]. Metabolites are small molecules (molecular weight <1500 Da) that change in the metabolic processes. Metabolomics is the science of studying the types, quantities, and patterns of change of metabolites after an organism is perturbed (such as genetic changes or environmental changes). The comprehensive testing of metabolites enables people to study the interaction between genes and the environment [9]. According to the purpose of testing and the diversity of samples, metabolomics can be divided into targeted metabolomics, untargeted metabolomics and so on. Targeted metabolomics is the qualitative and quantitative analysis of specific metabolites or types in samples by using standards. This method has higher reproducibility and sensitivity, but needs to purchase standards for methodological validation. Untargeted metabolomics is a comprehensive approach to identifying all small molecule metabolites within a certain range in a biological sample. It helps to discover new biomarkers [10,11]. Recent studies in metabolomics investigations involve the analysis of blood, urine, hair and other samples. Among these, urine collection is non-invasive and lacks the homeostatic mechanisms present in blood, making it more likely to magnify exercise-induced metabolic changes [12,13].

Developing an exercise prescription involves determining the mode, intensity, duration and frequency of exercise. These same factors also influence the changes in metabolites caused by exercise. Current exercise metabolomics studies concentrate on the changes following prolonged, high-intensity, single mode of exercise, but fewer studies focused on different modes of acute exercise [14–16].

## 2. Methods

### 2.1. Study participants

The protocol of this study was approved by the ethics committee of Shanghai University of Sport, China (NO. 102772021RT139). The recruitment period started from 1st February, 2022–30th June, 2022. Ten healthy male university students volunteered to participate in this study and they all signed a written informed consent. Participants aged 18–24, with a body mass index BMI, BMI = weight (kg)/ [height (m)]² ranging from 18.5 to 28, and who reported no injuries or illnesses in the past six months were included. Current smokers (individuals who reported using any tobacco product within the past 30 days) [17], heavy alcohol consumers (≥40 grams/day in men) [18], those with dietary preferences, those with a history of chronic diseases such as cardiovascular disease and family genetics, and those who take any medication or nutritional supplement that affects metabolism on a regular basis will be excluded. Participants' anthropometric characteristics were recorded in Table 1.

**Table 1. Participants' anthropometric characteristics.**

| Parameter | Value |
|---|---|
| Age (yrs) | 20.1±0.6 |
| Height (cm) | 173.4±3.9 |
| Weight (kg) | 66.6±4.1 |
| BMI (kg/m²) | 22.1±1.2 |

## 2.2. Exercise protocol

This study consisted of two exercise sessions, moderate-intensity aerobic exercise and moderate-intensity resistance exercise. Exercise intensity was estimated as a percentage of maximum heart rate (HRmax) (HRmax = 220 – age). Firstly, participants jogged two laps of a 400-meter track and walked one lap, keeping their average heart rate at 60–70% HRmax for 45 min. After more than 48h of rest, participants performed movements with a 5 kg barbell, 15 repetitions of each movement × 3 sets, with a 60s of rest between sets, keeping their average heart rate at 60–70% HRmax for 45 min [19]. All participants completed two exercise sessions. The participants' diets and work schedules were consistent. Within 24 hours before and after each exercise session, all participants were required to: 1) abstain from alcohol and caffeine intake; 2) consume standardised dinners and breakfasts. Furthermore, they did not engage in any moderate-to-vigorous physical activity during the experimental period.

## 2.3. Sample collection

Urine samples were collected 30 minutes before exercise, 30 minutes after exercise and 24 hours after recovery. Samples were aliquoted and stored at −80 °C until analysis.

## 2.4. Metabolite extraction

Urine samples stored at −80°C were thawed at room temperature. 80μl supernatant of every sample was taken in a mass spectrometry injection vial after pre-treatment. Blank samples were prepared by Milli-Q ultrapure water and quality control (QC) samples were prepared by mixing 10μl of each sample. One QC was inserted in every 10 samples for evaluating the system stability during the experiment.

## 2.5. LC-MS/MS analysis

For untargeted metabolomic analysis, UPLC-QTOF-MS analysis was performed on a Triple TOF 5600 mass spectrometer (SCIEX, USA) coupled to a Agilent 1290 Infinity II Ultra Performance Liquid Chromatography system (Agilent, USA) in both positive and negative ion modes. The separation was performed using an Acquity UPLC BEH Amide column (100 × 2.1 mm, 1.7 μm, Waters, USA) in binary gradient mode. Column temperatures was set to 40 °C and the injection volume used was 5 μl. Samples were placed in randomized order. One QC sample and one blank vial were run after each of the 10 urine samples.

## 2.6. Data analysis

Peak alignment, retention time correction, and peak area extraction were performed using XCMS software. Metabolite identification was performed using the MetDNA web platform (http://metdna.zhulab.cn/). Data were checked for completeness and normalized using the MetaboAnalyst web platform (https://www.metaboanalyst.ca/). Using multivariate statistical analysis to reduce the data dimensionality and help to identify significant variables between different samples. In particular, Orthogonal projections to latent structures-discriminant analysis (OPLS-DA) is more suitable for the separation of

two groups of samples than other multivariate statistical analysis methods by incorporating orthogonal signal correction (OSC), which is capable of filtering information not related to the subgroups while concentrating relevant information in one principal component, thus maximizing the display of differences between groups. Supervised analytical methods are prone to overfitting. A permutation test with 1000 repetitions is used to validate the model's reliability. The trends and significance of metabolite changes were presented in a volcano plot. Differential metabolites were screened by the mean normalized peak intensity fold change between the two groups (FC), the p-value of a two-tailed t-test on peak intensities, and the variable importance in projection obtained from the OPLS-DA analysis (VIP) (FC < 0.5 or FC > 2, p < 0.05, and VIP > 1). The screened differential metabolites were then imported into MetaboAnalyst platform for metabolic pathway analysis, including pathway analysis and enrichment analysis. Fig 1 shows the workflow for this study.

## 3. Results

### 3.1. Comparison of metabolic characteristics before and immediately after aerobic exercise

**3.1.1. Multivariate statistical analysis.** OPLS-DA plot (Fig 2a) shows that samples were significantly separated between the two groups. After 1,000 repetitions of permutation test (Fig 2b), the model fits well.

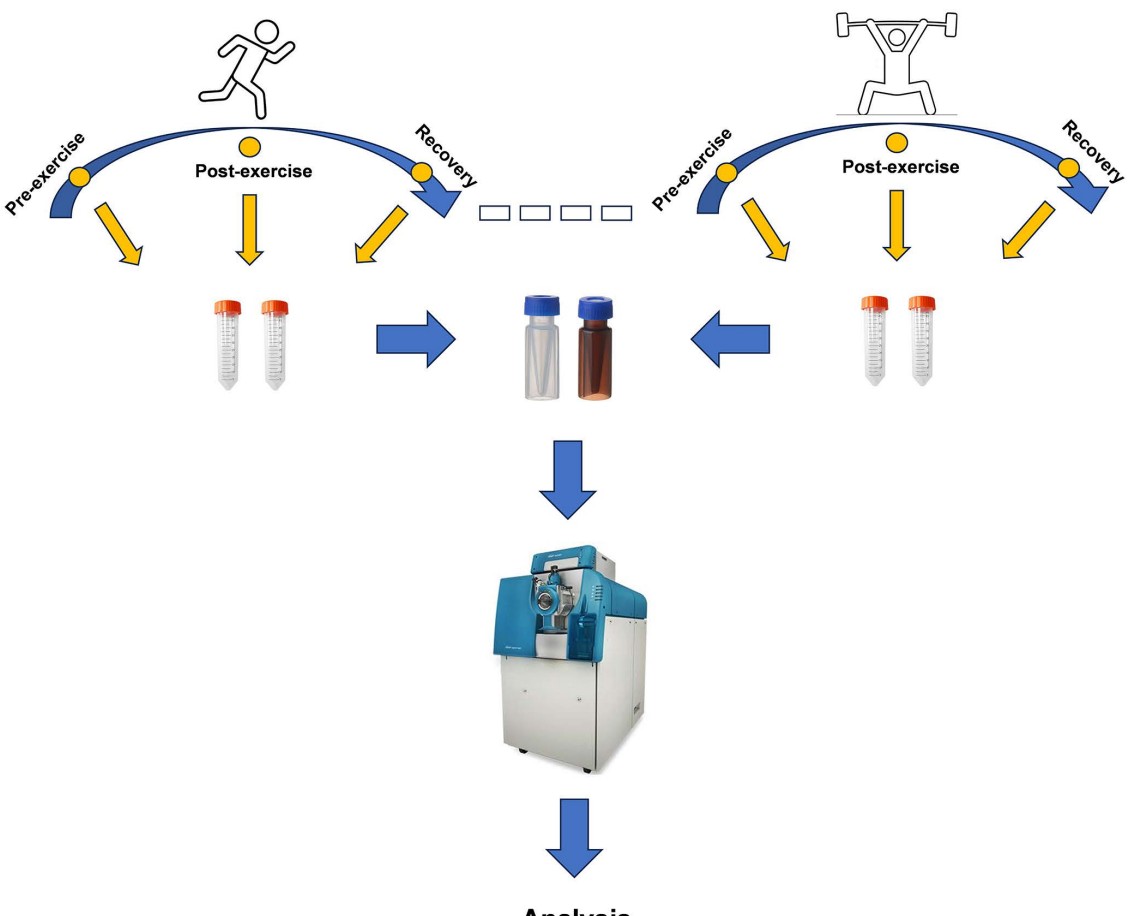

**Fig 1. Workflow for metabolomics profile analysis of urine samples based on mass spectrometry (MS).**

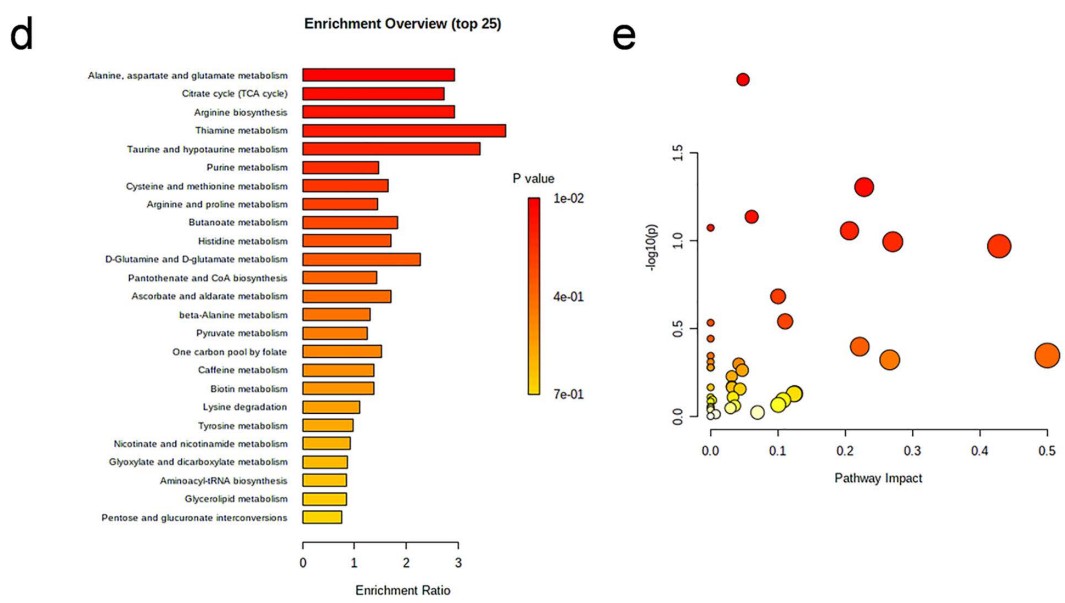

**Fig 2. Metabolic profile changes immediately after aerobic exercise.** (a) OPLS-DA plot (before exercise vs. immediate post-exercise). (b) The result of 1,000 repetitions of permutation test. (c) Volcano plot of differential metabolites. (d) Enrichment analysis of differential pathways. (e) Topology analysis of differential pathways.

**3.1.2. Differential metabolites screening.** The volcano plot (Fig 2c) indicates the distribution of changes in metabolites between the two groups (before and immediately after aerobic exercise). A total of 29 differential metabolites were screened, including 10 up-regulated and 19 down-regulated, listed in Table 2.

**3.1.3. Metabolic pathway analysis.** Enrichment analysis (Fig 2d) and pathway analysis (Fig 2e) were performed on the screened differential metabolites. The differential pathways are listed in Table 3 ($p < 0.05$ and Impact> 0). Alanine, aspartate and glutamate metabolism, and the tricarboxylic acid cycle are differential metabolic pathways.

### 3.2. Comparison of metabolic characteristics immediately and 24h after aerobic exercise

**3.2.1. Multivariate statistical analysis.** The OPLS-DA plot (Fig 3a) shows the significantly separation between the two groups. After 1,000 repetitions of permutation test (Fig 3b), the model fits well.

**3.2.2. Differential metabolites screening.** Volcano plot (Fig 3c) shows the distribution of metabolite changes immediately and 24 hours after aerobic exercise. A total of 22 differential metabolites were screened, including 18 up-regulated and 4 down-regulated ($p < 0.05$, FC<0.5 or FC>2, VIP>1), listed in Table 4.

**3.2.3. Metabolic pathway analysis.** After enrichment analysis (Fig 3d) and pathway analysis (Fig 3e), the metabolic pathway that showed significant change is listed in Table 5 ($p < 0.05$ and Impact>0). Histidine metabolism is the differential metabolic pathway.

### 3.3. Comparison of metabolic characteristics before and immediately after resistance exercise

**3.3.1. Multivariate statistical analysis.** The OPLS-DA plot (Fig 4a) shows the significantly separation between the two groups. After 1,000 repetitions of permutation test (Fig 4b), the model fits well.

**3.3.2. Differential metabolites screening.** Volcano plot (Fig 4c) shows the distribution of metabolite changes before and immediately after resistance exercise. A total of 38 differential metabolites were screened, including 16 up-regulated and 22 down-regulated ($p < 0.05$, FC<0.5 or FC>2, VIP>1), listed in Table 6.

**3.3.3 Metabolic pathway analysis.** After enrichment analysis (Fig 4d) and pathway analysis (Fig 4e), the metabolic pathways that showed significant changes are listed in Table 7 ($p < 0.05$ and Impact> 0). Alanine, aspartate and glutamate metabolism, arginine biosynthesis, D-glutamine and D-glutamate metabolism are the differential metabolic pathways.

### 3.4. Comparison of metabolic characteristics immediately and 24h after resistance exercise

**3.4.1. Multivariate statistical analysis.** The OPLS-DA plot (Fig 5a) shows the significantly separation between the two groups. After 1,000 repetitions of permutation test (Fig 5b), the model fits well.

**3.4.2. Differential metabolites screening.** Volcano plots (Fig 5c) show the distribution of metabolite changes immediately and 24 hours after resistance exercise. A total of 29 differential metabolites were screened, including 28 up-regulated and 1 down-regulated ($p < 0.05$, FC<0.5 or FC>2, VIP>1), listed in Table 8.

**3.4.3. Metabolic pathway analysis.** After enrichment analysis (Fig 5d) and pathway analysis (Fig 5e), the metabolic pathways that showed significant changes are listed in Table 9 ($p < 0.05$ and Impact> 0). Tryptophan metabolism, alanine, aspartate and glutamate metabolism, histidine metabolism are the differential metabolic pathways.

**Table 2. Differential metabolites before and immediately after aerobic exercise.**

| KEGG | HMDB | mz | Retention Time (RT, s) | Compound Name | FC | FDR | VIP | Trend |
|---|---|---|---|---|---|---|---|---|
| C00279 | HMDB0001321 | 199 | 682.2 | D-Erythrose 4-phosphate | 0.17 | 0.00419 | 1.48 | ↓ |
| C00093 | HMDB0000126 | 173.02 | 642.4 | Glycerophosphoric acid | 0.19 | 0.01369 | 1.32 | ↓ |
| C01081 | HMDB0002666 | 365.05 | 676.7 | Thiamin monophosphate | 0.2 | 0.00755 | 1.43 | ↓ |
| C00021 | HMDB0000939 | 385.13 | 659.7 | S-Adenosyl-L-homocysteine | 0.21 | 0.01369 | 1.27 | ↓ |
| C01620 | HMDB0000943 | 135.03 | 631.8 | Threonate | 0.22 | 0.02705 | 1.19 | ↓ |
| C05526 | METPA0607 | 425.08 | 888.7 | S-Glutathionyl-L-cysteine | 0.22 | 0.01844 | 1.25 | ↓ |
| C05844 | HMDB0004195 | 291.01 | 645.9 | 5-L-Glutamyl-taurine | 0.24 | 0.01247 | 1.31 | ↓ |
| C01187 | HMDB0244292 | 256.09 | 602.8 | Ketodeoxyoctonate | 0.25 | 0.01247 | 1.36 | ↓ |
| C05730 | METPA0653 | 435.24 | 548.1 | Glutathionylspermidine | 0.27 | 0.01844 | 1.26 | ↓ |
| C00262 | HMDB0000157 | 137.05 | 240.1 | Hypoxanthine | 0.28 | 0.02037 | 1.25 | ↓ |
| C04462 | HMDB0012266 | 307.12 | 315.1 | N-Succinyl-2-L-amino-6-oxoheptanedioate | 0.28 | 0.02037 | 1.24 | ↓ |
| C20775 | N/A | 360.03 | 570 | β-Citryl-L-glutamate | 0.29 | 0.00239 | 1.62 | ↓ |
| C00022 | HMDB0000243 | 147.03 | 258.3 | Pyruvate | 0.34 | 0.04207 | 1.15 | ↓ |
| C01103 | HMDB0000218 | 369.04 | 621 | Orotidine 5'-phosphate | 0.37 | 0.01369 | 1.32 | ↓ |
| C00362 | HMDB0001044 | 346.05 | 511.9 | dGMP | 0.38 | 0.04484 | 1.05 | ↓ |
| C03539 | METPA0405 | 285.12 | 448.7 | S-Ribosyl-L-homocysteine | 0.41 | 0.01891 | 1.28 | ↓ |
| C00378 | HMDB0000235 | 304.07 | 569.6 | Thiamine | 0.42 | 0.00449 | 1.49 | ↓ |
| C00077 | HMDB0000214 | 115.09 | 455.5 | L-Ornithine | 0.43 | 0.04830 | 1.07 | ↓ |
| C15532 | HMDB0000856 | 218.11 | 491.8 | N-Acetyl-L-citrulline | 0.46 | 0.03864 | 1.14 | ↓ |
| C00135 | HMDB0000177 | 156.08 | 784.8 | L-Histidine | 2.17 | 0.01032 | 1.32 | ↑ |
| C00170 | HMDB0001173 | 298.1 | 81.8 | 5'-Methylthioadenosine | 2.19 | 0.04542 | 1.04 | ↑ |
| C00047 | HMDB0000182 | 145.1 | 634.1 | L-Lysine | 2.51 | 0.00426 | 1.47 | ↑ |
| C05824 | HMDB0000731 | 201.98 | 513.3 | S-Sulfo-L-cysteine | 2.53 | 0.00478 | 1.44 | ↑ |
| C05340 | HMDB0060441 | 246.16 | 566.2 | β-Alanyl-L-arginine | 2.6 | 0.00239 | 1.58 | ↑ |
| C02427 | HMDB0000679 | 188.1 | 634.2 | L-Homocitrulline | 2.64 | 0.00351 | 1.5 | ↑ |
| C00158 | HMDB0000094 | 191.02 | 732.6 | Citrate | 2.84 | 0.00449 | 1.53 | ↑ |
| C01674 | HMDB0003556 | 407.16 | 586.3 | Chitobiose | 3.01 | 0.00373 | 1.5 | ↑ |
| C05598 | HMDB0000821 | 211.11 | 101.7 | Phenylacetylglycine | 3.32 | 0.01247 | 1.34 | ↑ |
| C01035 | HMDB0003464 | 166.06 | 657.8 | 4-Guanidinobutanoate | 6.75 | 0.00823 | 1.37 | ↑ |

**Table 3. Differential pathways before and immediately after aerobic exercise.**

| No. | Pathway | Hits/Total | p | Impact |
|---|---|---|---|---|
| 1 | Alanine, aspartate and glutamate metabolism | 6/28 | 0.01208 | 0.04808 |
| 2 | Tricarboxylic acid cycle (TAC) | 4/20 | 0.04941 | 0.22801 |

## 4. Discussion

This study investigated how different time points and exercise modes influence the metabolic characteristics of human urine. Significant differences in metabolic profiles were observed between pre-exercise and immediately post-exercise conditions. In contrast, changes 24 hours after exercise were less pronounced, suggesting that the body had not fully

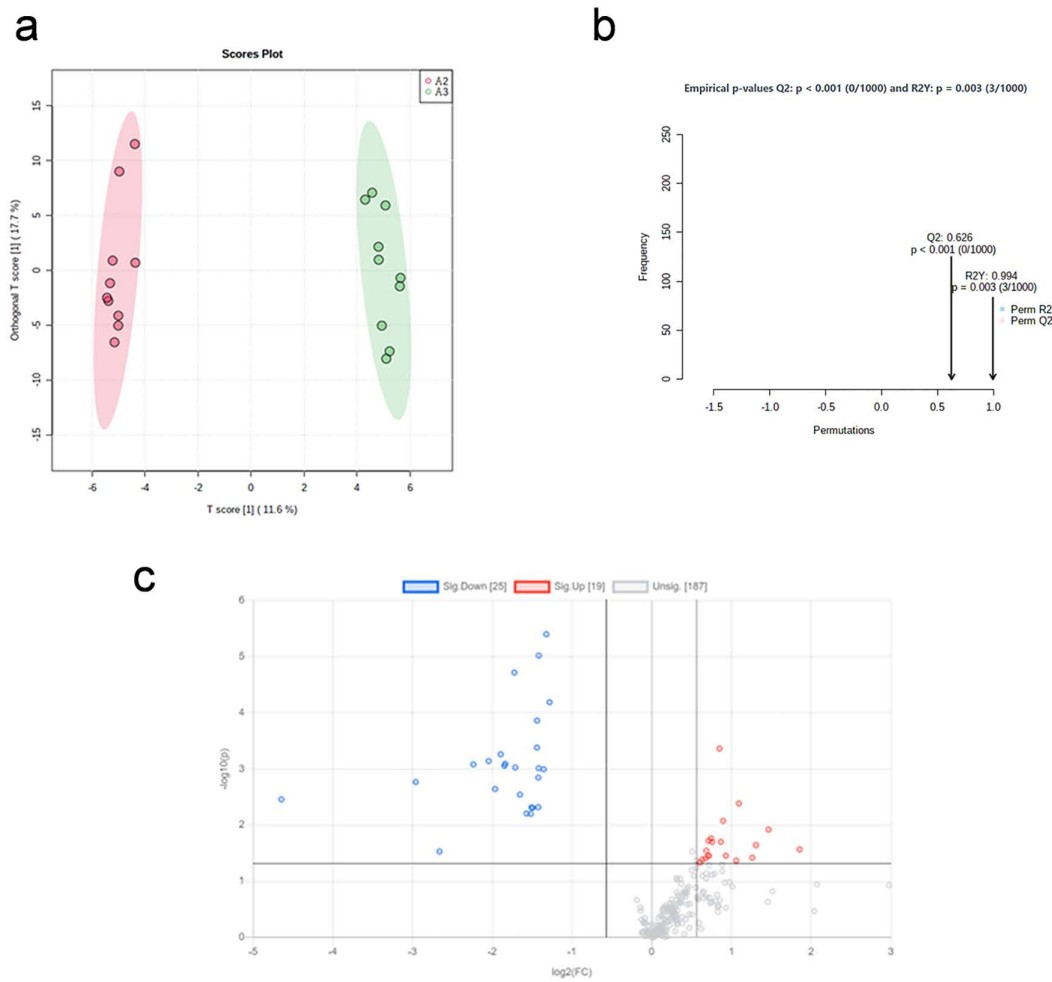

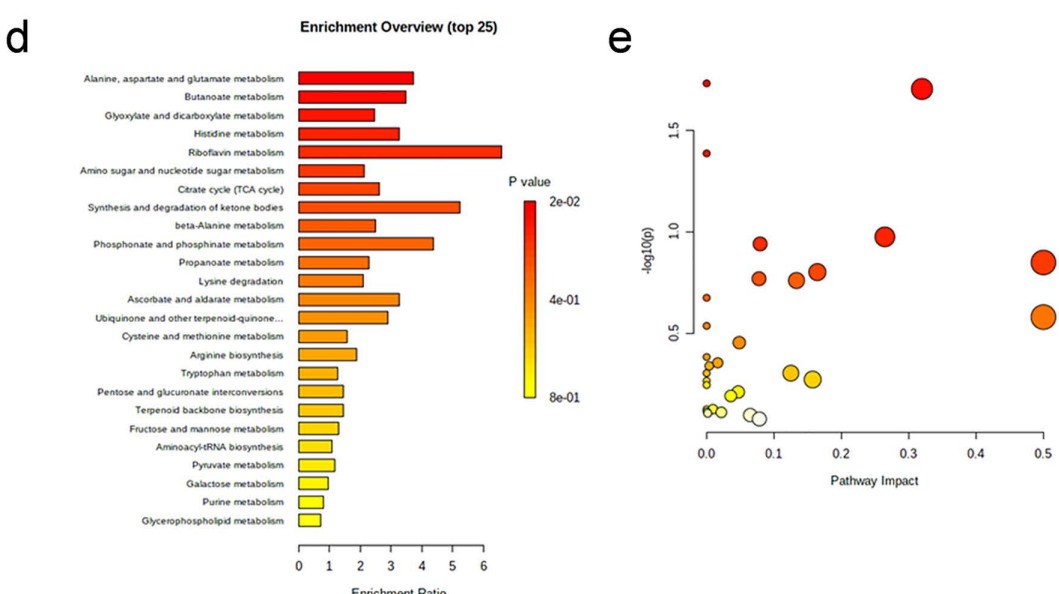

**Fig 3. Metabolic profile changes recovery 24h after aerobic exercise.** (a) OPLS-DA plot (immediate post-exercise vs. 24h post-exercise). (b) The result of 1,000 repetitions of permutation test. (c) Volcano plot of differential metabolites. (d) Enrichment analysis of differential pathways. (e) Topology analysis of differential pathways.

**Table 4. Differential metabolites immediately and 24h after aerobic exercise.**

| KEGG | HMDB | mz | RT | Compound Name | FC | FDR | VIP | Trend |
|---|---|---|---|---|---|---|---|---|
| C04037 | METPA0457 | 272.99 | 748.5 | D-Galacturonate 1-phosphate | 0.3 | 0.01562 | 2.3 | ↓ |
| C19872 | N/A | 167.09 | 266.7 | FAMP | 0.42 | 0.01562 | 2.42 | ↓ |
| C01104 | HMDB0000925 | 98.06 | 469.3 | Trimethylamine N-oxide | 0.45 | 0.01562 | 2.18 | ↓ |
| C05298 | HMDB0000343 | 285.15 | 380.9 | 2-Hydroxyestrone | 0.48 | 0.04719 | 2.09 | ↓ |
| C02427 | HMDB0000679 | 212.1 | 542.2 | L-Homocitrulline | 2.1 | 0.02962 | 1.14 | ↑ |
| C00559 | HMDB0000101 | 234.1 | 225.9 | Deoxyadenosine | 2.13 | 0.02326 | 1.24 | ↑ |
| C00575 | HMDB0000058 | 330.06 | 584.9 | Cyclic AMP | 2.16 | 0.01562 | 1.02 | ↑ |
| C16671 | N/A | 143.08 | 210.9 | 3,6-Dihydronicotinic acid | 2.19 | 0.04301 | 1.13 | ↑ |
| C15532 | HMDB0000856 | 218.11 | 491.8 | N-Acetyl-L-citrulline | 2.2 | 0.00635 | 1.27 | ↑ |
| C05340 | HMDB0060441 | 268.13 | 795.6 | β-Alanyl-L-arginine | 2.22 | 0.02059 | 1.01 | ↑ |
| C00402 | HMDB0006483 | 132.03 | 685.9 | D-Aspartate | 2.25 | 0.04719 | 1.26 | ↑ |
| C05933 | HMDB0004224 | 213.11 | 187.6 | N(omega)-Hydroxyarginine | 2.31 | 0.04089 | 1.02 | ↑ |
| C00942 | HMDB0001314 | 344.04 | 571.8 | Cyclic GMP | 2.37 | 0.04719 | 1.17 | ↑ |
| C20579 | N/A | 303.11 | 378 | Cyclopeptine | 2.41 | 0.03477 | 1.07 | ↑ |
| C04462 | HMDB0012266 | 307.12 | 315.1 | N-Succinyl-2-L-amino-6-oxoheptanedioate | 2.48 | 0.03477 | 1.13 | ↑ |
| C00544 | HMDB0000130 | 184.06 | 257 | Homogentisate | 2.59 | 0.02855 | 1.01 | ↑ |
| C04137 | N/A | 305.15 | 848.4 | Octopine | 2.59 | 0.01903 | 1.31 | ↑ |
| C00588 | HMDB0001565 | 202.11 | 373.7 | Choline phosphate | 2.74 | 0.00839 | 1.47 | ↑ |
| C00311 | HMDB0000193 | 173.01 | 521.8 | Isocitrate | 2.76 | 0.01982 | 1.49 | ↑ |
| C05401 | HMDB0006790 | 277.09 | 491.3 | Galactosylglycerol | 2.94 | 0.01426 | 1.66 | ↑ |
| C00826 | HMDB0304400 | 250.06 | 542.7 | L-Arogenate | 2.95 | 0.02266 | 1.55 | ↑ |
| C20775 | N/A | 360.03 | 570 | β-Citryl-L-glutamate | 3.04 | 0.04521 | 2.15 | ↑ |
| C00262 | HMDB0000157 | 137.05 | 240.1 | Hypoxanthine | 3.27 | FDR | 1.39 | ↑ |
| C05951 | HMDB0003080 | 477.23 | 361.7 | Leukotriene D4 | 3.35 | 0.01562 | 1.74 | ↑ |

**Table 5. Differential pathways immediately and 24h after aerobic exercise.**

| No. | Pathway | Hits/Total | p | Impact |
|---|---|---|---|---|
| 1 | Histidine metabolism | 3/16 | 0.01969 | 0.31967 |

returned to its pre-exercise state by this time point. Several amino acids (including aspartic acid, glutamic acid, histidine and tryptophan, etc.) exhibited significant changes in concentration before and after exercise. Amino acid metabolism was identified as a key finding in the metabolic pathway analysis. Although significant changes in the same metabolic pathways may occur before and after two sessions of exercise, the metabolites involved and their trends are different. Considering the interconnected nature of metabolic pathways, certain metabolites are involved in multiple pathways concurrently, thereby requiring a comprehensive consideration of the reasons.

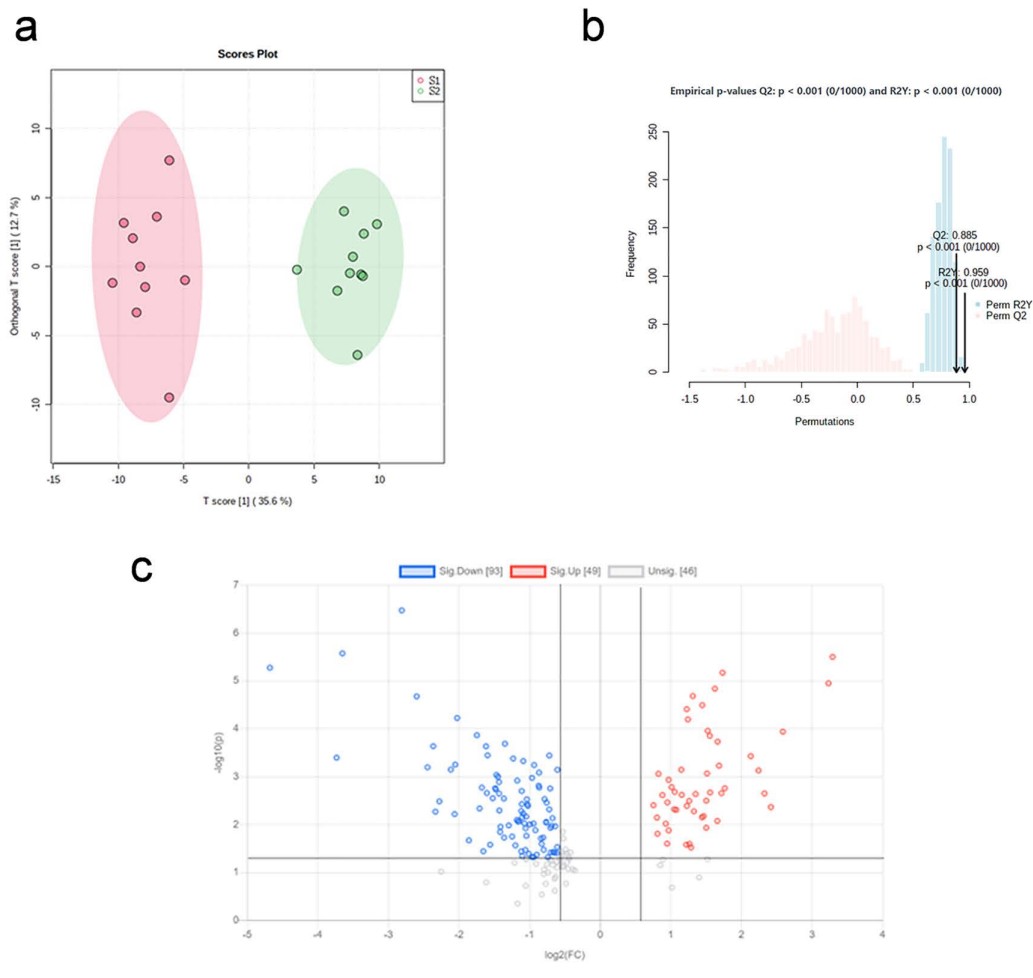

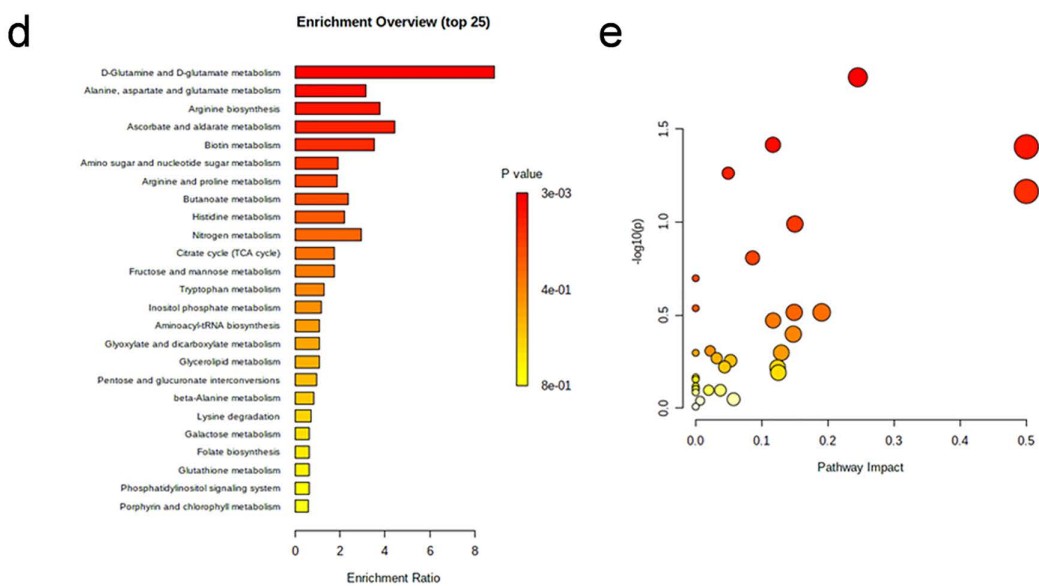

**Fig 4. Metabolic profile changes immediately after resistance exercise.** (a) OPLS-DA plot (before vs. immediate post-exercise). (b) The result of 1,000 repetitions of permutation test. (c) Volcano plot of differential metabolites. (d) Enrichment analysis of differential pathways. (e) Topology analysis of differential pathways.

**Table 6. Differential metabolites before and immediately after resistance exercise.**

| KEGG | HMDB | mz | RT | Compound Name | FC | FDR | VIP | Trend |
|---|---|---|---|---|---|---|---|---|
| C02888 | HMDB0006797 | 243.02 | 655.3 | Sorbose 1-phosphate | 0.03 | 0.00025 | 1.36 | ↓ |
| C01233 | HMDB0000114 | 238.04 | 653.9 | sn-Glycero-3-phosphoethanolamine | 0.06 | 0.00300 | 1.15 | ↓ |
| C00108 | HMDB0001123 | 136.04 | 238.5 | Anthranilate | 0.07 | 0.00020 | 1.48 | ↓ |
| C00262 | HMDB0000157 | 135.03 | 234.7 | Hypoxanthine | 0.13 | 0.00629 | 1.47 | ↓ |
| C05835 | METPA0699 | 230.05 | 201.1 | 2-Formaminobenzoylacetate | 0.17 | 0.00437 | 1.37 | ↓ |
| C04462 | HMDB0012266 | 307.12 | 315.1 | N-Succinyl-2-L-amino-6-oxoheptanedioate | 0.23 | 0.00920 | 1.37 | ↓ |
| C00093 | HMDB0000126 | 173.02 | 642.4 | Glycerophosphoric acid | 0.29 | 0.00711 | 1.02 | ↓ |
| C00366 | HMDB0000289 | 167.02 | 916.4 | Urate | 0.29 | 0.00647 | 1.17 | ↓ |
| C04037 | METPA0457 | 272.99 | 637 | D-Galacturonate 1-phosphate | 0.3 | 0.00647 | 1.02 | ↓ |
| C06104 | HMDB0000448 | 164.09 | 658.4 | Adipate | 0.31 | 0.04691 | 1.16 | ↓ |
| C02985 | HMDB0001265 | 243.03 | 657.3 | L-Fucose 1-phosphate | 0.32 | 0.00836 | 1.1 | ↓ |
| C03752 | HMDB0250755 | 254.08 | 622.7 | D-Glucosaminate | 0.32 | 0.00207 | 1.27 | ↓ |
| C03459 | METPA0398 | 207.02 | 631.8 | Oxaloglycolate | 0.34 | 0.00550 | 1.12 | ↓ |
| C01620 | HMDB0000943 | 135.03 | 631.8 | Threonate | 0.36 | 0.00837 | 1.08 | ↓ |
| C00019 | HMDB0001185 | 399.14 | 826.8 | S-Adenosyl-L-methionine | 0.38 | 0.00300 | 1.15 | ↓ |
| C00559 | HMDB0000101 | 269.14 | 314.1 | Deoxyadenosine | 0.38 | 0.00202 | 1.24 | ↓ |
| C04874 | HMDB0002275 | 256.1 | 548.1 | Dihydroneopterin | 0.39 | 0.00525 | 1.05 | ↓ |
| C02779 | N/A | 237.06 | 387.2 | 2-Dehydro-D-glucose | 0.41 | 0.01661 | 1.01 | ↓ |
| C03872 | HMDB0031950 | 287.07 | 606.1 | L-Serine-phosphoethanolamine | 0.42 | 0.00702 | 1.12 | ↓ |
| C00026 | HMDB0000208 | 164.06 | 257.8 | 2-Oxoglutarate | 0.43 | 0.00327 | 1.13 | ↓ |
| C16069 | METPA1235 | 229 | 631.6 | 3-Sulfolactate | 0.44 | 0.00999 | 1.09 | ↓ |
| C01367 | HMDB0003540 | 406.08 | 688 | 3'-AMP | 0.49 | 0.00368 | 1.17 | ↓ |
| C01674 | HMDB0003556 | 407.16 | 586.3 | Chitobiose | 2.21 | 0.00999 | 1.08 | ↑ |
| C00025 | HMDB0000148 | 168.03 | 641 | L-Glutamate | 2.22 | 0.00920 | 1.32 | ↑ |
| C01152 | HMDB0000001 | 168.08 | 635.9 | N(pi)-Methyl-L-histidine | 2.24 | 0.00662 | 1.39 | ↑ |
| C00398 | HMDB0000303 | 159.09 | 147.4 | Tryptamine | 2.25 | 0.00720 | 1.03 | ↑ |
| C02091 | METPA0247 | 265.09 | 630.8 | (S)-Ureidoglycine | 2.33 | 0.00437 | 1.43 | ↑ |
| C02946 | HMDB0003681 | 144.07 | 135.5 | 4-Acetamidobutanoate | 2.39 | 0.00711 | 1.11 | ↑ |
| C00191 | HMDB0000127 | 193.04 | 646.6 | D-Glucuronate | 2.43 | 0.00060 | 1.41 | ↑ |
| C02728 | HMDB0002038 | 161.14 | 469.9 | N6-Methyl-L-lysine | 2.58 | 0.00429 | 1.25 | ↑ |
| C01100 | HMDB0304403 | 242.04 | 642.9 | L-Histidinol phosphate | 2.66 | 0.01540 | 1.3 | ↑ |
| C00047 | HMDB0000182 | 145.1 | 634.1 | L-Lysine | 2.94 | 0.00389 | 1.4 | ↑ |
| C02427 | HMDB0000679 | 188.1 | 634.2 | L-Homocitrulline | 3.18 | 0.00252 | 1.43 | ↑ |
| C01031 | HMDB0001550 | 372.02 | 506.1 | S-Formylglutathione | 3.21 | 0.00711 | 1.21 | ↑ |
| C01035 | HMDB0003464 | 166.06 | 657.8 | 4-Guanidinobutanoate | 3.94 | 0.02905 | 1.32 | ↑ |
| C02991 | METPA0345 | 198.99 | 117.6 | L-Rhamnono-1,4-lactone | 4.36 | 0.00711 | 1.17 | ↑ |
| C02700 | HMDB0060485 | 235.07 | 369.2 | L-Formylkynurenine | 5.1 | 0.01433 | 1.29 | ↑ |
| C05939 | HMDB0038516 | 282.1 | 820.2 | Linatine | 8.05 | 0.00351 | 1.4 | ↑ |

**Table 7. Differential pathways before and immediately after resistance exercise.**

| No. | Pathway | Hits/Total | p | Impact |
|---|---|---|---|---|
| 1 | Alanine, aspartate and glutamate metabolism | 5/28 | 0.0167 | 0.2452 |
| 2 | Arginine biosynthesis | 3/14 | 0.03849 | 0.11675 |
| 3 | D-Glutamine and D-glutamate metabolism | 2/6 | 0.03945 | 0.5 |

Fig 6 compares the trends of metabolites in alanine, aspartate and glutamate metabolism pathway before and immediately after aerobic and resistance exercise. Specifically, L-asparagine levels decreased immediately after aerobic exercise but recovered 24 hours later, coinciding with an increase in D-asparagine levels. In contrast, L-asparagine levels showed a significant increase immediately following resistance exercise. Aspartate is converted to oxaloacetate under the catalysis of aspartate aminotransferase, entering the tricarboxylic acid cycle to generate energy. During aerobic exercise, L-asparagine is rapidly mobilized and converted into aspartic acid by deamidation. This process replenishes intermediates in TAC to meet energy demands [20]. At the same time, aspartic acid is also an important precursor for gluconeogenesis. Additionally, aspartate facilitates the delivery of potassium and magnesium ions to the myocardium, thereby enhancing myocardial contractility, reducing oxygen consumption, and protecting coronary circulation under hypoxic conditions [21]. An increase in D-asparagine may act as a regulatory signal to inhibit the excessive degradation of L-asparagine or promote its synthesis, thereby helping to maintain metabolic homeostasis of asparagine in the body [22].

Resistance exercise is one of the most effective stimulators of the mTOR pathway, a central regulator of cellular growth and protein synthesis. Studies have shown that asparagine itself can activate mTORC1 signaling and may serve as a substitute, particularly when intracellular leucine levels are inadequate [23]. Therefore, the elevation of L-asparagine following resistance exercise may contribute to the body's anabolic environment, promoting protein translation in preparation for muscle repair and growth. Muscle micro-damage induced by resistance exercise represents a form of cellular stress. Asparagine has been shown to play a critical role in the endoplasmic reticulum stress response, aiding in proper protein folding [24,25]. Therefore, the increase in L-asparagine levels may represent a protective adaptation to exercise-induced cellular stress. This transforms asparagine from a common "building material" into a key "metabolic signal".

Fig 7 shows the trends of metabolites in D-Glutamine and D-glutamate metabolism pathway before and immediately after resistance exercise. L-glutamate level increase immediate after exercise while α-ketoglutarate level decrease immediate after exercise. These changes can be attributed to exercise-induced micro-damage to muscle fibers, particularly during the eccentric phase, which accelerates muscle protein breakdown and releases amino acids, including glutamate. Furthermore, α-ketoglutarate is diverted into the TAC to meet the heightened energy demands. Glutamate is synthesized in body from α-ketoglutarate and alanine or other amino acids by transamination, which is a major process in amino acid catabolism. Studies have shown that overtraining leads to a significant rise in blood glutamate levels and that the rise correlates with exercise intensity [26]. During aerobic exercise, the metabolism of materials and the supply of energy remain relatively stable and continuous, without the characteristic peak in glutamate levels [27]. Within 24 hours of exercise, the body enters a recovery and growth phase, which is primarily focused on protein synthesis. The L-glutamate level subsequently declines. Glutamate is converted into L-glutamine, which is in high demand for protein synthesis and as a fuel for immune and intestinal cells. This sustained synthesis consumes glutamate pools [28]. One study showed that skeletal muscle glutamine level decreased significantly following a single heavy exercise session, and post-exercise glutamine supplementation was effective in maintaining muscle size and normal immune function [29]. The glutamate-glutamine cycle is crucial for ammonia transport. In muscles and the brain, glutamine is synthesized from glutamate and ammonia; it is then transported to the liver and kidneys, where it is hydrolyzed back to glutamate, safely disposing of ammonia [30]. Additionally, glutamate is decarboxylated to form γ-aminobutyric acid (GABA), an inhibitory neurotransmitter in the central

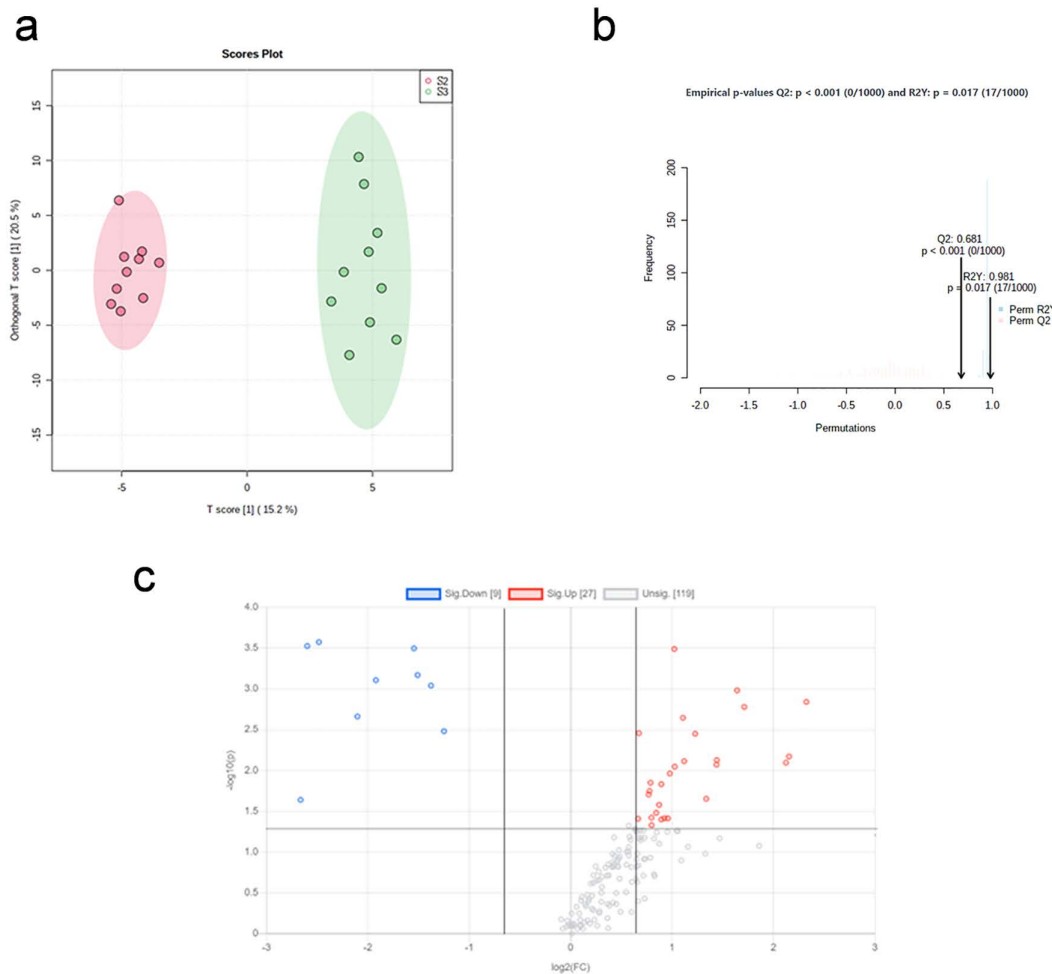

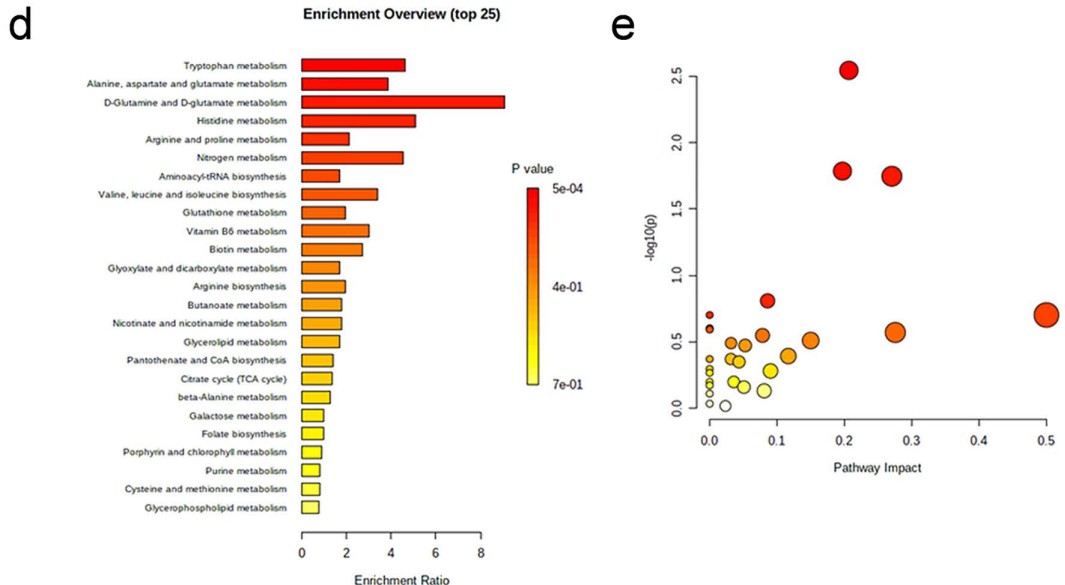

**Fig 5. Metabolic profile changes recovery 24h after resistance exercise.** (a) OPLS-DA plot (immediate post-exercise vs. 24h post-exercise). (b) The result of 1,000 repetitions of permutation test. (c) Volcano plot of differential metabolites. (d) Enrichment analysis of differential pathways. (e) Topology analysis of differential pathways.

**Table 8. Differential metabolites immediately and 24h after resistance exercise.**

| KEGG | HMDB | mz | RT | Compound Name | FC | FDR | VIP | Trend |
|---|---|---|---|---|---|---|---|---|
| C03287 | HMDB0001228 | 228.02 | 462.4 | L-Glutamyl 5-phosphate | 0.34 | 0.01257 | 1.83 | ↓ |
| C03296 | METPA0384 | 297.11 | 630.5 | N2-Succinyl-L-arginine | 2.14 | 0.03828 | 1.12 | ↑ |
| C06114 | HMDB0060477 | 222.08 | 760.1 | γ-Glutamyl-3-aminopropiononitrile | 2.33 | 0.01764 | 1.09 | ↑ |
| C04593 | HMDB0006471 | 207.05 | 385.8 | Methylisocitrate | 2.38 | 0.01682 | 1.28 | ↑ |
| C00093 | HMDB0000126 | 173.02 | 642.4 | Glycerophosphoric acid | 2.44 | 0.04018 | 1.02 | ↑ |
| C00879 | HMDB0000639 | 209.03 | 691.9 | D-Galactarate | 2.49 | 0.04898 | 1.18 | ↑ |
| C00534 | HMDB0001431 | 227.1 | 590.1 | Pyridoxamine | 2.5 | 0.03768 | 1.28 | ↑ |
| C01743 | N/A | 360.12 | 835.3 | Paromamine | 2.52 | 0.04686 | 1.14 | ↑ |
| C20775 | N/A | 360.03 | 570 | β-Citryl-L-glutamate | 2.53 | 0.03668 | 1.53 | ↑ |
| C04020 | N/A | 241.12 | 725.7 | D-Lysopine | 2.54 | 0.17682 | 1.02 | ↑ |
| C05556 | METPA0614 | 364.15 | 547.4 | delta-(L-2-Aminoadipyl)-L-cysteinyl-D-valine | 2.56 | 0.01898 | 1.21 | ↑ |
| C05921 | HMDB0004220 | 591.18 | 335.6 | Biotinyl-5'-AMP | 2.56 | 0.03515 | 1.31 | ↑ |
| C16427 | HMDB0304396 | 336.16 | 296 | Isopentenyl adenosine | 2.63 | 0.03108 | 1.12 | ↑ |
| C04874 | HMDB0002275 | 256.1 | 548.1 | Dihydroneopterin | 2.68 | 0.03515 | 1.41 | ↑ |
| C00439 | HMDB0000854 | 233.08 | 731.9 | N-Formimino-L-glutamate | 3.06 | 0.01707 | 1.52 | ↑ |
| C05515 | METPA0604 | 229.05 | 392.9 | 5-Ureido-4-imidazole carboxylate | 3.09 | 0.01257 | 1.25 | ↑ |
| C00826 | HMDB0304400 | 250.06 | 542.7 | L-Arogenate | 3.12 | 0.03862 | 1.9 | ↑ |
| C00954 | HMDB0000197 | 234.08 | 731.8 | Indole-3-acetate | 3.22 | 0.04682 | 1.57 | ↑ |
| C00182 | HMDB0000757 | 684.25 | 871.9 | Glycogen | 3.23 | 0.03629 | 1.09 | ↑ |
| C05570 | HMDB0003045 | 248.13 | 568 | Ergothioneine | 3.25 | 0.04561 | 1.5 | ↑ |
| C03090 | HMDB0001128 | 245.06 | 629.2 | 5-Phosphoribosylamine | 3.5 | 0.04682 | 1.58 | ↑ |
| C00455 | HMDB0000229 | 333.06 | 731.4 | Nicotinamide D-ribonucleotide | 4.12 | 0.03668 | 1.52 | ↑ |
| C15700 | METPA1197 | 255.07 | 397.1 | γ-Glutamyl-γ-aminobutyraldehyde | 4.32 | 0.04561 | 1.55 | ↑ |
| C00019 | HMDB0001185 | 379.11 | 731.4 | S-Adenosyl-L-methionine | 4.33 | 0.02020 | 1.61 | ↑ |
| C00262 | HMDB0000157 | 135.03 | 234.7 | Hypoxanthine | 5.25 | 0.03682 | 1.79 | ↑ |
| C00108 | HMDB0001123 | 136.04 | 238.5 | Anthranilate | 7.92 | FDR | 1.78 | ↑ |

**Table 9. Differential pathways immediately and 24h after resistance exercise.**

| No. | Pathway | Hits/Total | p | Impact |
|---|---|---|---|---|
| 1 | Tryptophan metabolism | 6/41 | 0.00286 | 0.20673 |
| 2 | Alanine, aspartate and glutamate metabolism | 4/28 | 0.0164 | 0.19712 |
| 3 | Histidine metabolism | 3/16 | 0.01791 | 0.27049 |

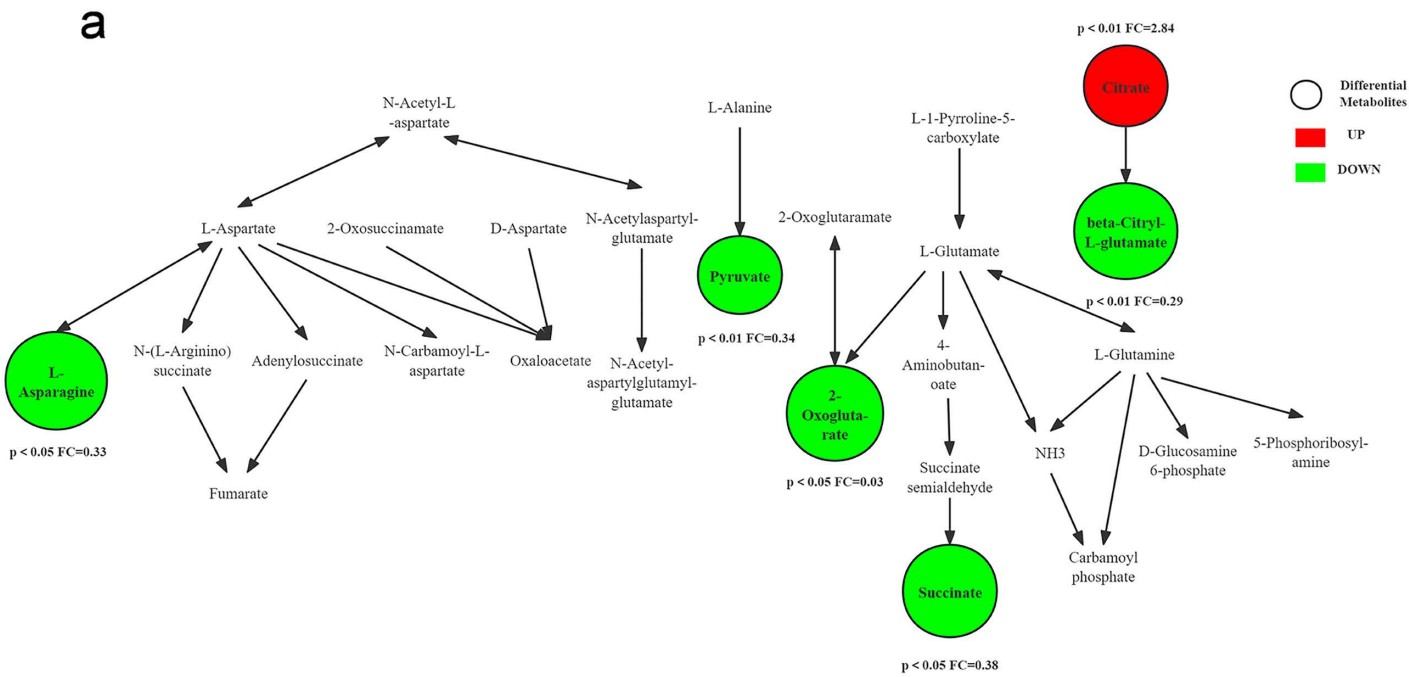

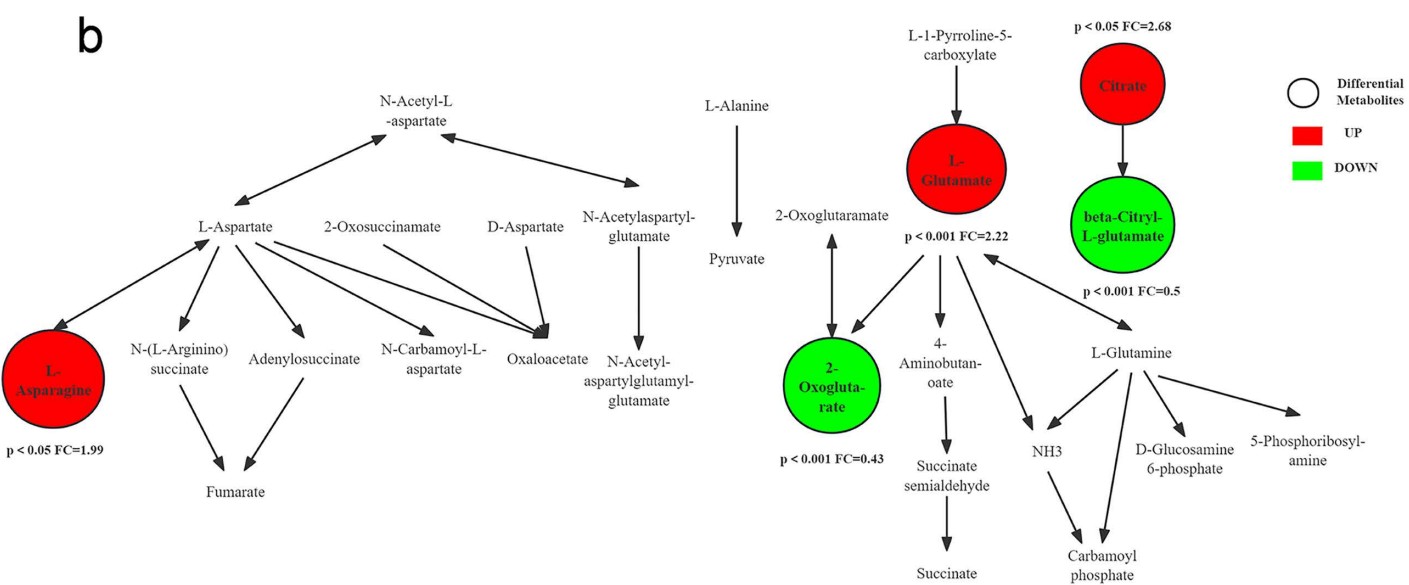

**Fig 6. Alanine, aspartate and glutamate metabolism pathway: differential metabolites changes.** (a) Aerobic exercise (before vs. immediate post-exercise). (b) Resistance exercise (before vs. immediate post-exercise).

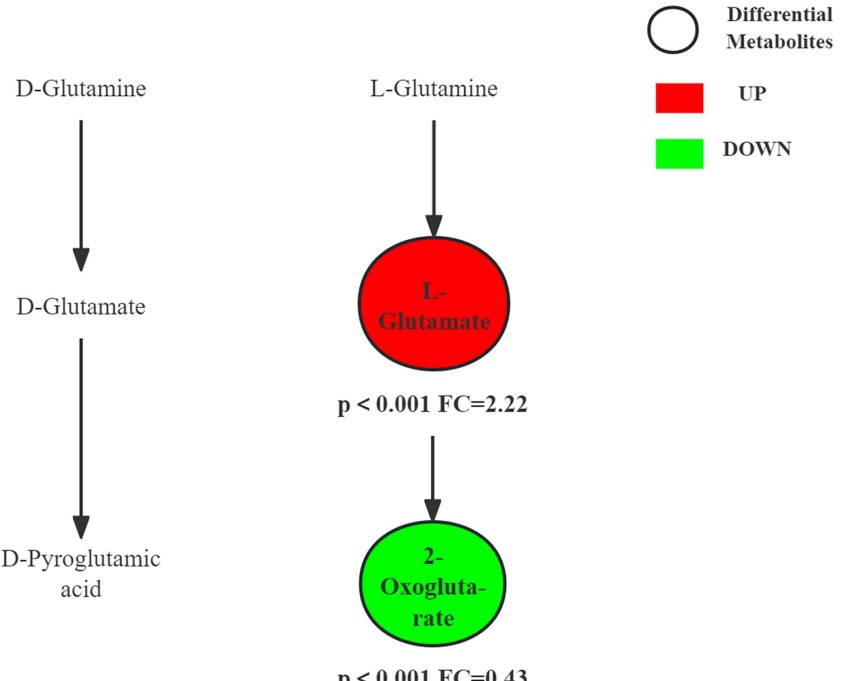

**Fig 7. D-Glutamine and D-glutamate metabolism pathway and changes in differential metabolites before and immediately after resistance exercise.**

nervous system. Levels of glutamate and GABA are often used as indicators of exercise-induced fatigue, with changes in their levels potentially reflecting protective mechanisms of the central nervous system to maintain neuronal activity [31].

Fig 8 compares the trends of metabolites in histidine metabolism pathway immediately and 24h after aerobic and resistance exercise. A significant decrease in L-histidine levels and a corresponding increase in its catabolic products were observed following aerobic exercise, indicating that histidine continues to be metabolized in the body 24 hours post-exercise. Conversely, L-histidine levels increased 24 hours after resistance exercise. The catabolic product of L-histidine, β-alanyl-L-histidine (also known as carnosine, CAR), is a dipeptide found in high concentrations in skeletal muscle. Both histidine and CAR have strong capacity for scavenging free radicals and antioxidants [32]. Studies have shown that type II muscle fibers have higher levels of CAR and/or muscle histidine dipeptide than type I muscle fibers, which is related to the energy transfer of CAR under anaerobic conditions and its ability to act as a pH buffer to reduce the concentration of hydrogen ions in the myocyte. Some athletes engaged in long-term resistance training have higher levels of CAR [33]. L-histidine decarboxylates to produce histamine, which is converted to imidazole acetaldehyde catalyzed by diamine oxidase. Histamine is closely associated with inflammatory and allergic reactions in body, and plays a crucial role in regulating muscle microcirculation during exercise and sustained vasodilation post exercise [34].

Fig 9 shows the trends of metabolites in tryptophan metabolism pathway 24 hours after resistance exercise. Tryptophan, a gluconeogenic and ketogenic amino acid, exhibited a significant increase in its metabolic intermediates, including L-kynurenine, indoleacetic acid, and 5-methoxyindoleacetic acid, which represent three distinct catabolic pathways for tryptophan. More than 90% of tryptophan is metabolized via the kynurenine pathway. Indoleamine 2,3-dioxygenase (IDO) and tryptophan 2,3-dioxygenase (TDO) are the rate-limiting enzymes initiating the kynurenine pathway. Resistance exercise causes microdamage to muscles, triggering the release of pro-inflammatory cytokines such as interferon-γ (IFN-γ), tumor necrosis factor-α (TNF-α), interleukin-1β (IL-1β) and interleukin-6 (IL-6). IFN-γ is a key inducer of IDO, and TNF-α and IL-1β

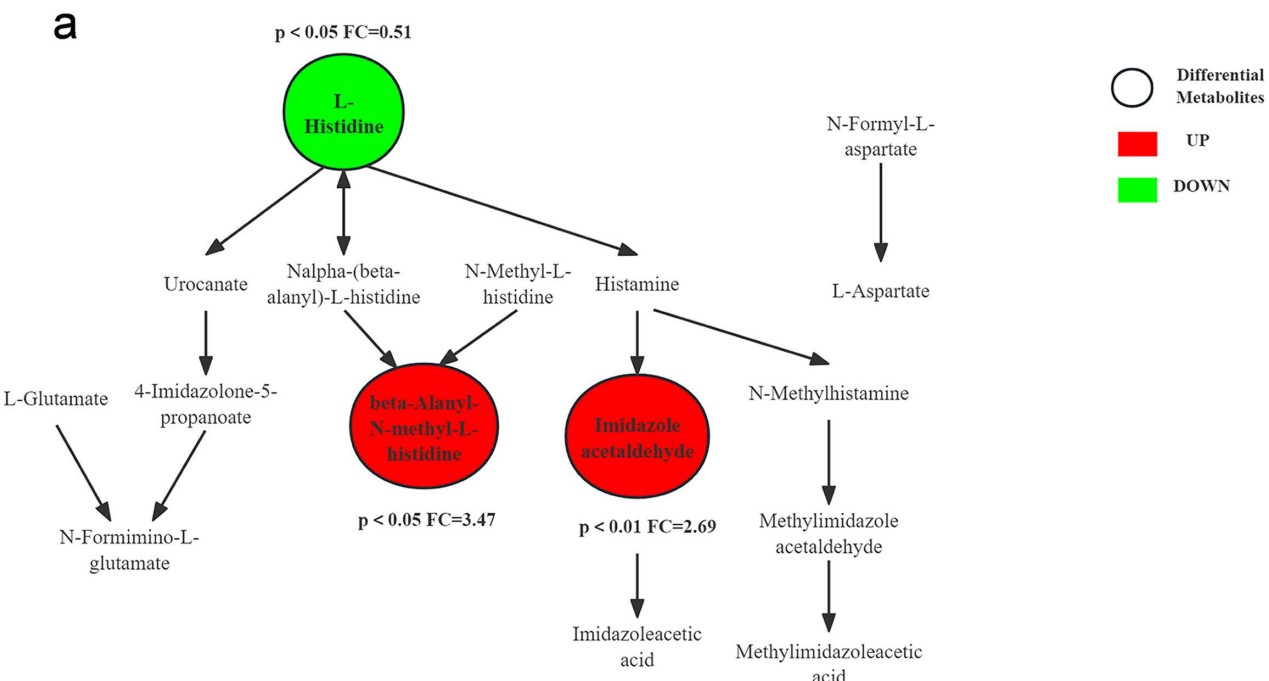

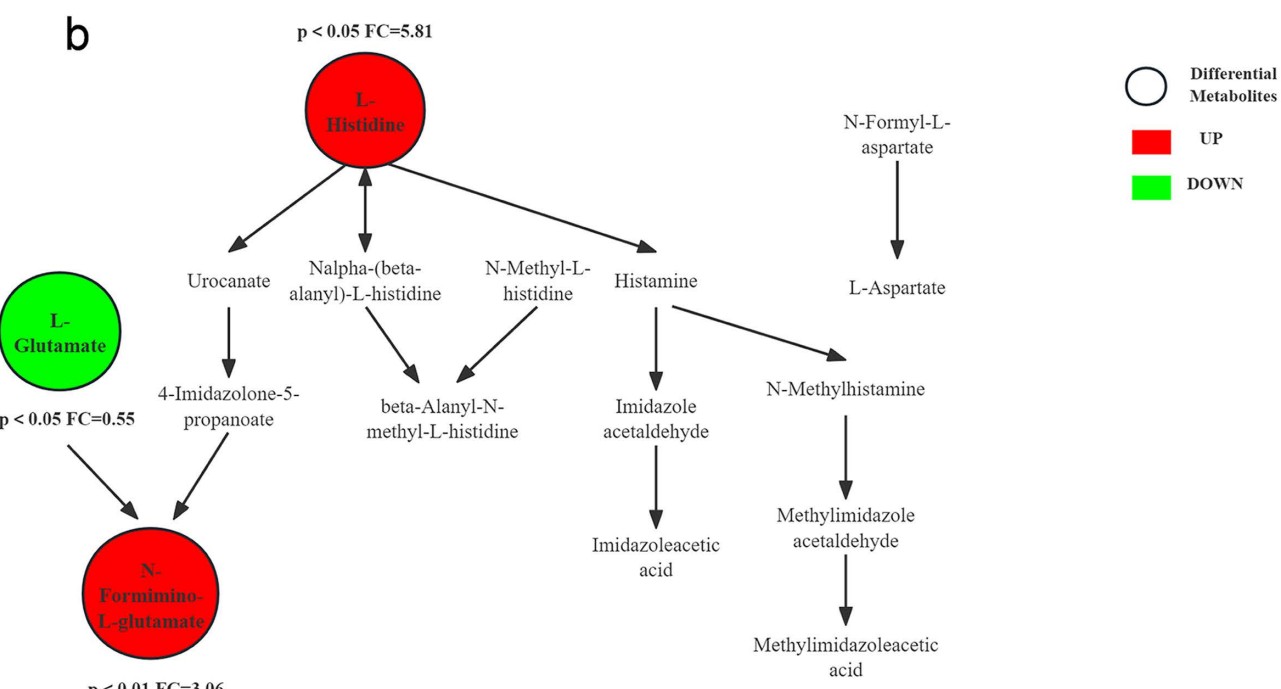

**Fig 8. Histidine metabolism pathway: differential metabolites changes.** (a) Aerobic exercise (immediate vs. 24h post-exercise). (b) Resistance exercise (immediate vs. 24h post-exercise).

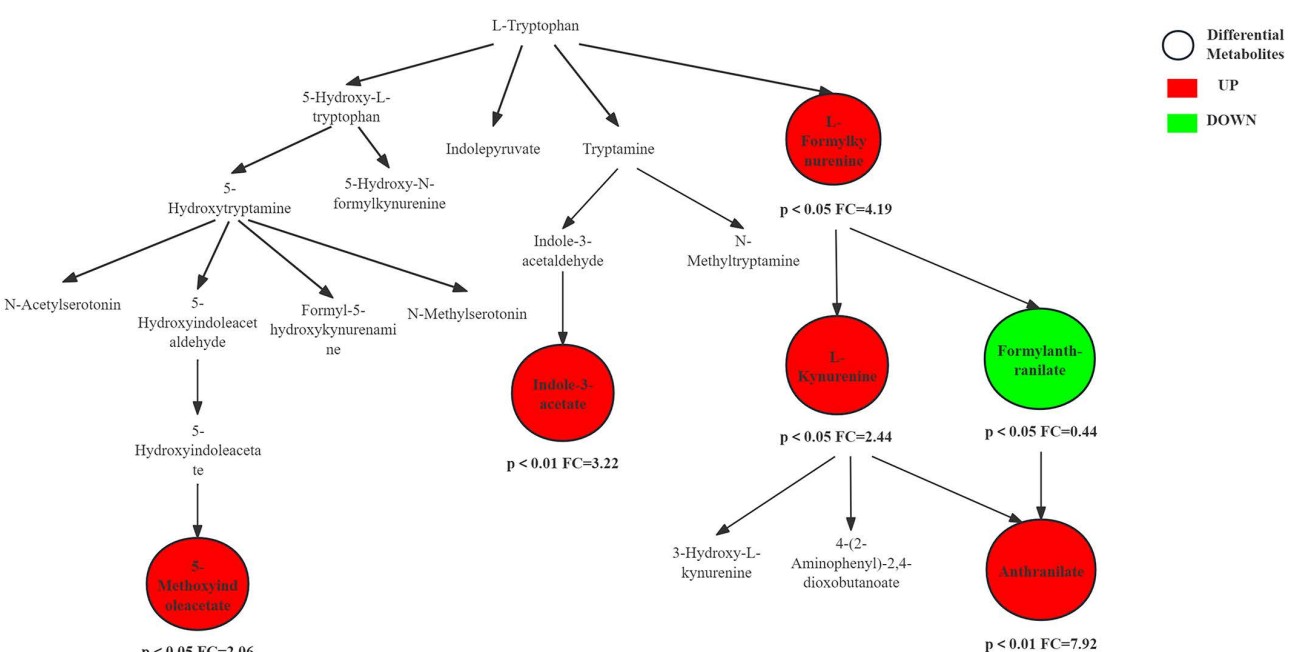

**Fig 9. Tryptophan metabolism pathway and changes in differential metabolites immediately and 24h after resistance exercise.**

synergistically enhance its expression [35–37]. Downstream metabolites of the kynurenine pathway, such as kynurenic acid, have immunosuppressive and antioxidant properties which help the body transition from a pro-inflammatory state to a reparative one [38]. Resistance exercise activates the hypothalamic–pituitary–adrenal axis, resulting in the release of cortisol. Glucocorticoids are potent inducers of TDO in the liver. Cortisol levels may remain elevated or be in the recovery process within 24 hours post-exercise. Cortisol triggers TDO, working together with inflammation factor-induced IDO to drive tryptophan breakdown along the kynurenine pathway [39]. The activation of the tryptophan pathway 24 hours after resistance exercise represents an adaptive response to tissue damage. Its primary function is to manage inflammation, promote immune tolerance and create an environment conducive to tissue repair via the kynurenine pathway, rather than merely providing energy [40]. Studies have shown that the tryptophan/kynurenine ratio or kynurenine levels in the blood may serve as potential biomarkers for evaluating athletes' training load, inflammation levels and recovery status [41].

## 5. Conclusion

Acute exercise induces significant alterations in the metabolic characteristics of human urine. Non-targeted metabolomic analysis based on LC-MS provides a more comprehensive and accurate reflection of the changes before and after exercise.

Two exercise modes promote exercise adaptation through specific amino acid metabolism. Aerobic exercise mobilizes aspartic acid, glutamic acid, and others to sustain a steady energy supply. In contrast, resistance exercise engages tryptophan and histidine, as well as their metabolites, to mediate immunosuppressive and antioxidant responses, thereby facilitating the transition from exercise-induced inflammation to a recovery state.

## 6. Limitation

The study had a relatively small sample size and significant variations in metabolic characteristics were observed among individuals. Furthermore, samples were only collected from healthy male college students at two time points following

a single exercise session, which may not be sufficient for describing the recovery process comprehensively. The study primarily focused on the molecular level and did not establish a correlation between exercise-induced changes in metabolites and functional indicators.

Future studies could expand sample sizes and increase sampling time points to achieve more accurate and comprehensive descriptions. Multi-omics approaches could be employed to cross-validate results and further refine biomarker selection. Comparative analyses could be conducted between individuals with different diseases or other populations and healthy males. Long-term exercise studies could also establish links between specific metabolic pathways and the effects of physical activity on organ function.

## Supporting information

**S1 File. Sample collection and pretreatment procedures, along with LC-MS conditions.**
(DOCX)

**S1 Fig. Shows the metabolomics total ion chromatogram (TIC).**
(TIF)

**S2 Fig. Shows the PCA plots.**
(TIF)

## Author contributions

**Formal analysis:** Junjie Kuang.

**Investigation:** Junjie Kuang.

**Methodology:** Xin Xu.

**Project administration:** Jie Ju.

**Resources:** Jie Ju.

**Supervision:** Xin Xu.

**Writing – original draft:** Junjie Kuang.

**Writing – review & editing:** Xin Xu.

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
