## [Decision Letter · Decision Letter 0]

20 Sep 2025

Dear Dr. Kuang,

Thank you for submitting your manuscript to PLOS ONE. After careful consideration, we feel that it has merit but does not fully meet PLOS ONE’s publication criteria as it currently stands. Therefore, we invite you to submit a revised version of the manuscript that addresses the points raised during the review process.

We look forward to receiving your revised manuscript.

Kind regards,

Saki Raheem, PhD

Academic Editor

PLOS ONE

Journal Requirements:

Reviewers' comments:

Reviewer's Responses to Questions

**Comments to the Author**

1. Is the manuscript technically sound, and do the data support the conclusions?

Reviewer #1: Yes

Reviewer #2: Yes

2. Has the statistical analysis been performed appropriately and rigorously?

Reviewer #1: Yes

Reviewer #2: No

3. Have the authors made all data underlying the findings in their manuscript fully available?

Reviewer #1: Yes

Reviewer #2: Yes

4. Is the manuscript presented in an intelligible fashion and written in standard English?

Reviewer #1: Yes

Reviewer #2: No

Reviewer #1: This manuscript presents an untargeted metabolomics investigation using LC-MS to explore the urinary metabolite profiles associated with single sessions of aerobic and resistance exercise in healthy male college students. The study attempts to characterize metabolic responses at multiple time points post-exercise and identify pathways potentially relevant to exercise physiology and personalized training prescriptions.

The topic is timely and contributes to the growing field of exercise metabolomics. However, several critical issues need to be addressed to improve the rigor, clarity, and interpretability of the manuscript.

1. Abstract, well summarized but slightly overstates implications (e.g., “novel basis for formulating exercise prescriptions”), Ethics and Data Sharing: Satisfactory, References: Appropriate and relevant, though a few key reviews on exercise metabolomics could be added.

2. The writing is mostly clear, but some sections (particularly Results and Discussion) are dense with data and weak in synthesis. Consider summarizing findings more succinctly, possibly in summary diagrams or bullet lists.

3. The Introduction is well written but could more clearly define what gap this study fills in current literature (e.g., few studies comparing acute aerobic vs. resistance responses using urine).

4. The study uses only 10 male participants, which significantly limits the statistical power and generalizability of the findings. While metabolomics studies often work with small cohorts, the manuscript should include a power analysis or justification for the sample size.

5. The all-male sample also precludes insights into sex-specific metabolic responses. This limitation must be clearly acknowledged and discussed.

6. The cross-over design lacks detail on randomization and whether the order of exercise modes (aerobic vs. resistance) was counterbalanced. Without this, the study is at risk of order or carryover effects.

7. The control over diet and other physical activity is mentioned but not quantified. A more rigorous dietary log, standardization, or assessment of energy balance would strengthen interpretation of metabolite fluctuations.

8. Urinary metabolomics is sensitive to dilution effects. There is no mention of normalization based on creatinine, specific gravity, or osmolality—standard practice to account for variation in urine concentration. This is a major omission that needs to be addressed for data reliability.

9. The study uses OPLS-DA, which is a supervised method prone to overfitting. There is no mention of model validation (e.g., cross-validation, permutation tests, R²/Q² values). These must be included to support the reliability of the findings.

10. The criteria for significance (FC, p, VIP) are standard, but multiple testing correction (e.g., FDR) is not reported. This is especially important in untargeted metabolomics with large feature sets.

11. While the pathway findings are interesting (e.g., glutamate, histidine, tryptophan metabolism), some interpretations stretch beyond the data. The causal links to fatigue, inflammation, or CNS activity are speculative and should be softened unless supported by additional measures (e.g., inflammatory markers, performance data).

12. The figures referenced (e.g., volcano plots, pathway enrichment) are essential, but some descriptions are redundant, and the captions lack detail (e.g., n per group, scale).

13. Some of the metabolite IDs in tables (e.g., KEGG IDs) could be supplemented with HMDB IDs or clearer biological roles for accessibility.

In summary, the authors should Include urine normalization methodology, report OPLS-DA validation and adjust for multiple comparisons Clarify design (randomization, counterbalancing), expand limitations (sample size, sex bias, speculative interpretation), tone down conclusions and strengthen discussion of how the findings align with or diverge from prior literature and improve figure clarity and labelling.

Reviewer #2: The study titled “Metabolomic Characteristics of Aerobic and Resistance Exercise Modes” by Junjie Kuang et.al explored significant changes in few metabolites such as amino acids in aerobic and resistance exercise persons using metabolomics approach.

The study makes an ordinary contribution to the understanding of the physical activity associated with metabolic regulations. This present research work is interesting and publishable after addressing some important concerns.

1. Novelty is limited. Author should explain exact hypothesis mechanism behind this study.

2. How this present study is translational to human pathophysiology and associated to mankind.

3. Why author explaining by focus on the amino acid metabolism instead of others like sungar and associated phosphate because these all are contributing factors in the muscle’s physiology.

4. Authors should represent the data accurately.

5. I would suggest to see some metabolomics articles that can be helpful for this study data representation.

6. The metabolites table is incorrectly represented i.e., RT. Please check carefully.

7. Author should provide the clearer figures and reasonable biology of the acquired data.

8. Author should provide the biology of metabolites associated with metabolic pathway or cycle that can be better for this theme of study.

9. Author should provide the more details of the study outcome in discussion section.

10. Elimination of these deficiencies can be understandable and publishable.

**Do you want your identity to be public for this peer review?** For information about this choice, including consent withdrawal, please see our Privacy Policy

Reviewer #1: No

Reviewer #2: **Yes: ** Ashutosh

---

## [Author Response · Author response to Decision Letter 1]

26 Oct 2025

Response to Reviewers

1 Summarizing the implications of the research as follows: This study aims to outline the metabolic responses induced by acute aerobic and resistance training in healthy male college students. It identifies indicators reflecting physiological status and exercise efficacy.

2 The innovative points of this study lie in: Collect urine samples and employe LC-MS-based metabolomic analysis to investigate the effects of different exercise modes and time points before and after a single exercise session on metabolic characteristics.

3 Hypothesis mechanism: Different types of acute exercise sessions elicit distinctly different urinary metabolome responses. Specifically, we anticipate that aerobic exercise will significantly alter levels of fatty acids and tricarboxylic acid cycle intermediates by activating mitochondrial oxidative function. In contrast, resistance exercise may lead to a sharp increase in lactate and purine metabolites through anaerobic glycolysis.

4

a) We fully acknowledge that the limited sample size (n=10) may impact statistical power. As a preliminary mechanistic exploratory study, our primary target is to screen key metabolites or pathways to generate hypotheses for follow-up large-scale confirmatory research. Many metabolomics studies have adopted similar sampling size strategies. (e.g. Floegel, A., et al. (2013), Barton, R. H., et al. (2010)). Through rigorous experimental controls and longitudinal sampling, the study has somewhat minimised confounding factors. We have reinforced this point in the “Limitations” section of the manuscript.

b) We fully acknowledge that the inability to elucidate gender-specific metabolic responses constitutes one of the core limitations of this study.

Our decision to include only male subjects was primarily driven by the need to control for substantial metabolic variability arising from hormonal fluctuations associated with gender and menstrual cycles. Such a design is both common and necessary during the initial stages of establishing methodological frameworks and exploring mechanisms within exercise metabolomics (e.g., Maher, A. D., et al. (2010), Reue, K., & Wende, A. R. (2021)). This approach aim to first delineate the core metabolic response profiles induced by aerobic and resistance exercise within a relatively homogeneous metabolic cohort. This establishes a reliable baseline for subsequent studies in more complex populations, such as females and different age groups.

We have reinforced this point in the “Limitations” section of the manuscript. We explicitly state that the present findings apply only to healthy young males. Furthermore, we call for future studies specifically addressing women and gender-related metabolic differences, which is considered a crucial step towards advancing personalised exercise prescription.

c) This study did not employ a crossover design for the sequence of exercises. It must be acknowledged that potential sequence effects and residual effects exist. We compare metabolic profiles before aerobic and resistance exercises. The PCA plots show the two groups largely overlapping, indicating minimal differences in metabolic characteristics between samples. Following a recovery period of 48h or more after aerobic exercise, human metabolism has largely returned to pre-exercise levels.

d) We acknowledge that the lack of quantitative monitoring of diet and daily activities constitutes another limitation of this study. However, we have adopted a standardised protocol to minimise its impact.

We have expanded the relevant description in the Methods section: Within 24 hours before and after each exercise session, all participants were required to: abstain from alcohol and caffeine intake; avoid any moderate-to-vigorous physical activity; and consume standardised dinners and breakfasts. These measures aim to minimise individual variability in pre-test metabolic baseline.

5 Urine concentration varies considerably between individuals. We fully acknowledge that correcting for concentration variability is essential for accurate data interpretation. However, due to sample limitations, we are unable to obtain additional urine samples for measuring creatinine or specific gravity. We recognise this as a limitation of the present study. To minimise the impact of urine concentration variation on results, we implement the following measures:

a) Standardised sample preparation: We adhere to a highly standardised sample preparation protocol, employing identical initial volumes and processing steps for all samples;

b) Data-driven normalisation: During data preprocessing, we apply total signal normalisation to correct for overall signal intensity differences between samples;

c) QC sample validation: QC samples are incorporated to monitor and ensure the stability of the entire analytical workflow. QC samples cluster tightly within the PCA plot.

Sample pretreatment procedures are listed in S1_File. S1_Fig1. shows the metabolomics total ion chromatogram (TIC). S1_Fig2. shows the PCA plots.

6 The results of the permutation test are shown in the figures. After 1,000 repetitions of permutation test, the models fit well.

7 Replace the p-values in the table with FDR adjusted p-values. Remove metabolites showing no significant change after correction.

8 Add HMDB IDs of the metabolites in the tables.

9 The descriptions for the figures and tables have been modified.

10 In this study, multiple amino acids (aspartic acid, glutamic acid, histidine, tryptophan, etc.) exhibit significant changes in levels before and after the two exercise modes, with metabolic pathway analysis results also focusing on amino acid metabolism. The discussion section mainly focuses on the differing trends in the same metabolite before and after two modes of exercise. Alternatively, the specific activation of a metabolite or pathway during one exercise section. The discussion is divided into four paragraphs:

a) The differing trends in changes in aspartic acid/asparagine before and after exercise in two modes;

b) Significant changes in glutamate/glutamine levels before and after resistance exercise;

c) The differing trends in histidine levels during the 24-hour recovery period following two modes of exercise;

d) Activation of the tryptophan metabolic pathway 24 hours after resistance exercise.

Speculative conclusions have been removed.

11 Acute exercise induces significant alterations in the metabolic characteristics of human urine. Two exercise modes promote exercise adaptation through specific amino acid metabolism. Aerobic exercise mobilizes aspartic acid, glutamic acid, and others to sustain a steady energy supply. In contrast, resistance exercise engages tryptophan and histidine, as well as their metabolites, to mediate immunosuppressive and antioxidant responses, thereby facilitating the transition from exercise-induced inflammation to a recovery state.

12 Future studies could expand sample sizes and increase sampling time points to achieve more accurate and comprehensive descriptions. Multi-omics approaches could be employed to cross-validate results and further refine biomarker selection. Comparative analyses could be conducted between individuals with different diseases or other populations and healthy males. Long-term exercise studies could also establish links between specific metabolic pathways and the effects of physical activity on organ function.

---

## [Decision Letter · Decision Letter 1]

28 Nov 2025

Metabolomic Characteristics of Aerobic and Resistance Exercise Modes

PONE-D-25-28714R1

Dear Dr. Kuang,

We’re pleased to inform you that your manuscript has been judged scientifically suitable for publication and will be formally accepted for publication once it meets all outstanding technical requirements.

Kind regards,

Saki Raheem, PhD

Academic Editor

PLOS ONE

Reviewers' comments:

Reviewer's Responses to Questions

**Comments to the Author**

Reviewer #1: All comments have been addressed

2. Is the manuscript technically sound, and do the data support the conclusions?

Reviewer #1: Yes

3. Has the statistical analysis been performed appropriately and rigorously?

Reviewer #1: Yes

4. Have the authors made all data underlying the findings in their manuscript fully available?

Reviewer #1: Yes

5. Is the manuscript presented in an intelligible fashion and written in standard English?

Reviewer #1: Yes

Reviewer #1: Thank you for carefully addressing all the comments and suggestions raised in the previous review. I have examined the revised version of your manuscript and found that all concerns have been adequately addressed. The revisions have improved the clarity and quality of the work.

I have no further comments or recommendations at this stage. The manuscript is acceptable for publication in its present form.

**Do you want your identity to be public for this peer review?** For information about this choice, including consent withdrawal, please see our Privacy Policy

Reviewer #1: **Yes: ** Rakesh Roshan Jha

---

## [Editor Report · Acceptance letter]

PONE-D-25-28714R1

PLOS ONE

Dear Dr. Kuang,

I'm pleased to inform you that your manuscript has been deemed suitable for publication in PLOS ONE. Congratulations! Your manuscript is now being handed over to our production team.

Kind regards,

on behalf of

Dr. Saki Raheem

Academic Editor

PLOS ONE